# SLA-v3: Spatial Linkability-Aware and Novelty-Encouraging State Heuristic for Exploration

## Abstract

Efficient exploration continues to be a pivotal challenge in reinforcement learning (RL), particularly in environments characterized by sparse rewards. While intrinsic motivation (IM) has proven effective for tackling hard exploration tasks, current IM approaches often struggle with the detachment-derailment (D-D) problem. This issue significantly curtails their effectiveness, especially in settings with extremely sparse rewards. Although methods like Go-Explore address D-D by explicitly archiving states to ensure revisitation, their dependency on state restoration limits their practical application in procedurally generated environments. In this paper, we argue that the root cause of the D-D problem lies in the underlying topological transition structure of the environment. Specifically, we observe that certain states become persistently difficult to traverse and revisit reliably when subjected to exploratory noise. To overcome this, we introduce a novel IM framework centered on state traversal difficulty. Within this framework, we propose the **S**patial **L**inkability-**A**ware **a**nd **N**ovelty-**E**ncouraging **S**tate **H**euristic (**SLAANESH**), abbreviated as **SLA-v3**. SLA-v3 tackles the D-D problem by utilizing the shortest-path quasi-metric from the initial state ($S_0$) as a heuristic for traversal difficulty. This mechanism generates sustainable exploratory incentives, particularly encouraging visit to hard-to-traverse states. Furthermore, SLA-v3 integrates a novelty detector, which serves to warm up the heuristic and effectively prevent stagnation in unproductive dead-end paths. Our extensive experimental evaluations on MiniGrid and challenging Atari environments (PitFall! and Montezuma's Revenge) robustly demonstrate the superior efficacy of SLA-v3.

## 1 Introduction

In reinforcement learning (RL), agents primarily rely on reward signals to optimize their policies. However, in sparse-reward environments, traditional algorithms Mnih et al. (2016; 2013); Schulman et al. (2017); Schrittwieser et al. (2020) frequently struggle to make meaningful policy updates. Instead, they often default to uniformly random exploration until a reward is accidentally encountered. This leads to a detrimental vicious cycle: uniformly random policies seldom discover sparse rewards, and without these critical signals, policies cannot effectively improve. Consequently, efficient exploration methods become indispensable for progress in such settings.

Intrinsic motivation (IM) has emerged as a prominent and successful framework for addressing the sparse reward problem in reinforcement learning Ladosz et al. (2022). This approach enhances the reward signal by leveraging historical experience replay, thereby facilitating more effective policy optimization. Contemporary high-performance IM methods are generally categorized into two distinct paradigms: **novelty**/curiosity-driven mechanisms and **diversity**-oriented approaches. Novelty-based methods Burda et al. (2018); Zhang et al. (2021b); Pathak et al. (2017); Pecháč et al. (2023); Guo et al. (2022); Zhang et al. (2021a) primarily incentivize the exploration of under-visited states, while diversity-oriented methods Henaff et al. (2022); Badia et al. (2020); Jiang et al. (2025); Wan et al. (2023); Raileanu & Rocktäschel (2020); Yuan et al. (2022) focus on enforcing diverse state visitation over shorter temporal horizons.

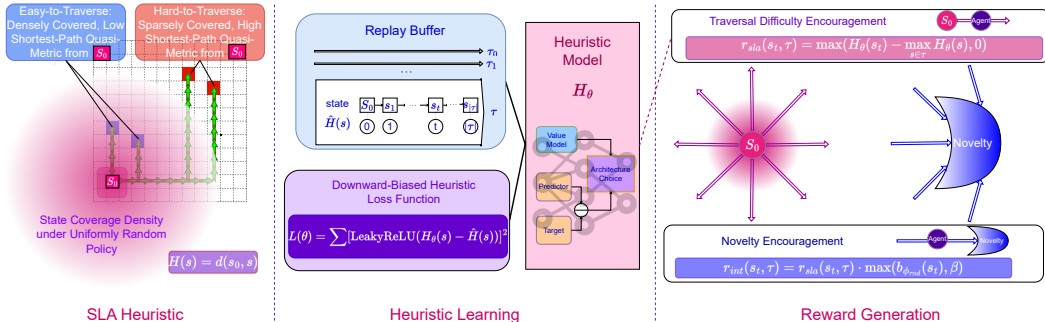

Figure 1: Overview of SLA-v3. (Left) SLA Heuristic: The *shortest-path quasi-metric* from the initial state $S_0$ serves as a traversal difficulty heuristic. States with higher quasi-metric values from $S_0$ are typically less frequently covered by a uniformly random exploratory policy. (Middle) The heuristic model, whose architecture can be a value model or a novelty detector, is optimized to predict the minimum episodic timestamp a state has been reached within the replay buffer. (Right) The intrinsic reward generated by SLA-v3 promotes visiting both hard-to-traverse states (distant from $S_0$) and novel states, guiding the agent away from well-explored initial regions towards high-novelty areas.

Despite their successes, a critical limitation of existing IM methods is their susceptibility to the Detachment-Derailment (D-D) problem Ecoffet et al. (2021), which significantly impedes exploration efficacy. More precisely, Derailment occurs when exploratory noise inadvertently disrupts the agent's ability to reliably visit potentially rewarding states. Detachment, on the other hand, manifests as a divergence between the actual state visitation space and the region deemed highly intrinsically motivating (i.e., the exploration frontier). Current state-of-the-art IM methods often fall short in adequately addressing this D-D problem: novelty-based approaches tend to suffer from diminishing exploration guidance and an exhaustible reward signal, whereas diversity-based methods prioritize exploration breadth at the expense of depth, often leading to suboptimal exploration strategies.

The Go-Explore framework Ecoffet et al. (2021) offers a compelling solution to the D-D problem by maintaining a comprehensive archive of all visited states. This mechanism guarantees state re-visitation and subsequent exploration, effectively circumventing the D-D issue through exhaustive state memorization. However, its practical applicability is severely constrained by its fundamental reliance on complete state recoverability, which is inherently unattainable in procedurally generated environments Küttler et al. (2020).

This context naturally leads to a fundamental question: Can we develop an exploration algorithm that effectively mitigates the D-D problem without imposing the strict requirements of state recovery? We assert that the environment's underlying topological transition structure is the root cause of the D-D problem. Our analysis reveals that states exhibit inherent asymmetric traversal difficulty: some states remain easily revisitable despite exploratory noise, while others necessitate sustained exploration incentives and minimal noise for reliable access.

Building upon this insight, we propose a novel intrinsic motivation paradigm grounded in state traversal difficulty. By utilizing temporal distance from the initial state $S_0$ as a robust traversal difficulty heuristic, we introduce the **S**patial **L**inkability-**A**ware **a**nd **N**ovelty-**E**ncouraging **S**tate **H**euristic (SLAANESH, abbreviated as SLA-v3), which is visually summarized in Figure 1. Our approach makes four key contributions: First, we formally define the shortest-path quasi-metric from $S_0$ as a heuristic metric to quantitatively assess detachment and derailment risk. Second, we develop a Go-Explore-inspired intrinsic reward mechanism that strategically guides the agent to **go** to the most hard-to-traverse known state and then **explore** extensively from that established frontier. Third, we integrate a novelty detector to facilitate heuristic warm-up and effectively prevent stagnation within local dead-end paths during a single trajectory. Fourth, we present extensive experimental results in MiniGrid and challenging Atari environments (specifically PitFall! and Montezuma's Revenge), which convincingly demonstrate the superior effectiveness of SLA-v3.

## 2 RELATED WORKS

### 2.1 INTRINSIC MOTIVATION

Intrinsic Motivation (IM) serves to augment sparse rewards by employing domain-agnostic heuristics and leveraging trajectory history. This field is currently advancing through three principal methodological approaches:

**Novelty-based** (or curiosity-driven/visit-count-based) IMs aim to promote exploration of under-visited states. They achieve this through various novelty quantification techniques, including density estimation Ostrovski et al. (2017); Bellemare et al. (2016), prediction variance Pathak et al. (2019), prediction error Burda et al. (2018); Pecháč et al. (2023); Pathak et al. (2017); Guo et al. (2022); Zhang et al. (2021b), variational objectives Zhang et al. (2021a), and information gain Kim et al. (2019); Bai et al. (2021); Kim et al. (2018); Houthooft et al. (2016); Mazzaglia et al. (2022). Nevertheless, novelty-based intrinsic motivation methods remain susceptible to the Detachment problem Ecoffet et al. (2021), where the exploration policy often fails to adequately cover intrinsically motivated states.

**Diversity-based** IMs are designed to promote short-term state space coverage across multiple temporal scales. These include transition-level approaches Raileanu & Rocktäschel (2020), episodic methods Badia et al. (2020); Henaff et al. (2022); Wan et al. (2023); Jiang et al. (2025), and inter-episodic strategies Yuan et al. (2022). Such approaches typically rely on state similarity metrics, ranging from simple hashing techniques Raileanu & Rocktäschel (2020); Zhang et al. (2021b); Flet-Berliac et al. (2021) to more advanced representation learning like inverse dynamics Pathak et al. (2017); Badia et al. (2020); Henaff et al. (2022); Raileanu & Rocktäschel (2020) and temporal distance Myers et al. (2024); Jiang et al. (2025). Inherently, diversity-based methods prioritize breadth over depth in exploration, potentially limiting their ability to effectively solve challenging exploration problems that demand deep state space traversal.

**Topology-Based** IMs exploit the environmental transition topology to guide exploration within sparse-connected transition structures. Key approaches in this category include spectral decomposition of state spaces Machado et al. (2017a;b); Klissarov & Machado (2023); Von Luxburg (2007) and successor/predecessor feature analysis Yu et al. (2024) for identifying bottlenecks. Our SLA-v3 framework also adopts the paradigm of topology modeling to identify and encourage the revisitation of hard-to-traverse states, specifically to counter the D-D problem.

### 2.2 GO-EXPLORE

The Go-Explore framework Ecoffet et al. (2021) represents the state-of-the-art for hard exploration problems. It has achieved unparalleled performance in notoriously challenging benchmarks, such as Montezuma's Revenge and PitFall! (though relying on sophisticated domain-specific knowledge). The efficacy of Go-Explore stems from two key mechanisms: (1) an exhaustive state archive that guarantees revisitation through either simulator-assisted state restoration or goal-conditioned policies, which effectively eliminates detachment by ensuring persistent access to promising states; and (2) archive-based state sampling, which circumvents derailment by resetting exploration from strategically selected initial states. Subsequent improvements to Go-Explore Höftmann et al. (2023); Gallouédec & Dellandréa (2023); Jia et al. (2024) have further enhanced its scalability, often through latent space archiving and density-based sampling techniques. However, Go-Explore's fundamental reliance on perfect state recoverability significantly limits its applicability to procedurally generated environments.

## 3 SLA-V3: SPATIAL LINKABILITY-AWARE AND NOVELTY-ENCOURAGING STATE HEURISTIC

### 3.1 SLA HEURISTIC: SHORTEST-PATH QUASI-METRIC FROM $S_0$

We propose a principled heuristic for state traversal difficulty that addresses two key requirements: (1) Detachment Risk Awareness, which quantifies the probability of state rediscovery under uniform

random policies when intrinsic rewards become uninformative; and (2) Derailment Risk Awareness, which measures susceptibility to exploratory noise during state visitation.

Our core heuristic, which we term the SLA Heuristic, is based on a shortest-path quasi-metric that quantifies the minimum number of temporal steps from the initial state $S_0$ to any given state $s$. It is formally expressed as:

$$H_{\text{sla}}(s) = d(s_0, s) = \min\{t \geq 0 : s_t = s\} \tag{1}$$

This $H_{\text{sla}}(s)$ value inherently serves as a measure of traversal difficulty. The choice of this quasi-metric is motivated by the observation that states requiring more sequential actions to reach from $S_0$ inherently exhibit greater sensitivity to perturbations from exploratory noise at each decision point. For singleton environments, $S_0$ directly corresponds to the episodic starting state. In procedurally generated environments, it represents the abstract initial configuration from which concrete environment instances are sampled before the first observation is generated.

## 3.2 Learning and Approximation of the SLA Heuristic

The ground truth shortest-path quasi-metric for each state is generally unavailable unless the true transition dynamics are known. Consequently, we approximate the heuristic using transition dynamic information derived from episodic trajectories within the replay buffer. For any state $s_t$ within an episodic trajectory with step count $t$, the ground truth shortest-path quasi-metric serves as an upper bound $t$, assuming a deterministic environment. Therefore, we use the episodic step count as the target for heuristic approximation and employ a downward-focused optimization strategy. This ensures the heuristic is optimized towards the minimum value observed across all trajectories.

$$\mathcal{L}(\theta) = \sum_{\tau \in \mathcal{D}} \sum_{s_t \in \tau} f_{\text{lrelu}} \left( H_\theta(s_t) - t \right)^2, \tag{2}$$

where $f_{\text{lrelu}}$ implements a leaky-ReLU Xu et al. (2015) style gradient weighting defined as:

$$f_{\text{lrelu}}(x) = \begin{cases} x & \text{if } x \geq 0, \\ \epsilon x & \text{if } x < 0, \end{cases} \tag{3}$$

with $0 < \epsilon \ll 1$ controlling the gradient scale.

Given that the approximator of the SLA heuristic (referred to as the *heuristic model* throughout the remainder of this paper) maps from the state space to a scalar, it is functionally equivalent to the value model used in PPO Schulman et al. (2017). Therefore, the neural architecture for our heuristic model can be directly determined by applying the same selection criteria as for the PPO value model, without necessitating neural architecture search or additional components (e.g., contrastive models or normalization schemes).

## 3.3 First Maximize, Then Explore

In each episode, the agent is designed to first **go** to the known hardest-to-traverse state and then **explore** from this established frontier. To achieve this behavior, SLA-v3 employs an implicit dual-phase reward mechanism: it initially encourages the agent to maximize the state heuristic value, followed by a phase of unguided exploration without additional intrinsic reward once the hardest-to-traverse frontier state has been reached. The heuristic component of the intrinsic reward is formally defined as:

$$r_{\text{sla}}(s_t, \tau) = \max \left( H_\theta(s_t) - \max_{i \in [1..t-1]} H_\theta(s_i), 0 \right), \tag{4}$$

where $\tau$ denotes the episodic trajectory and $H_\theta$ represents the heuristic model. This non-negativity constraint prevents performance degradation that could arise from heuristic underestimation, particularly for states that are indistinguishable (in a partial observability setting) or unexplored hard-to-traverse states.

## 3.4 Integration of Novelty Detector

We incorporate a novelty detector into SLA-v3 through two distinct mechanisms. First, it acts as a reward modulator to prevent exploration stagnation in local dead-end paths. The complete intrinsic

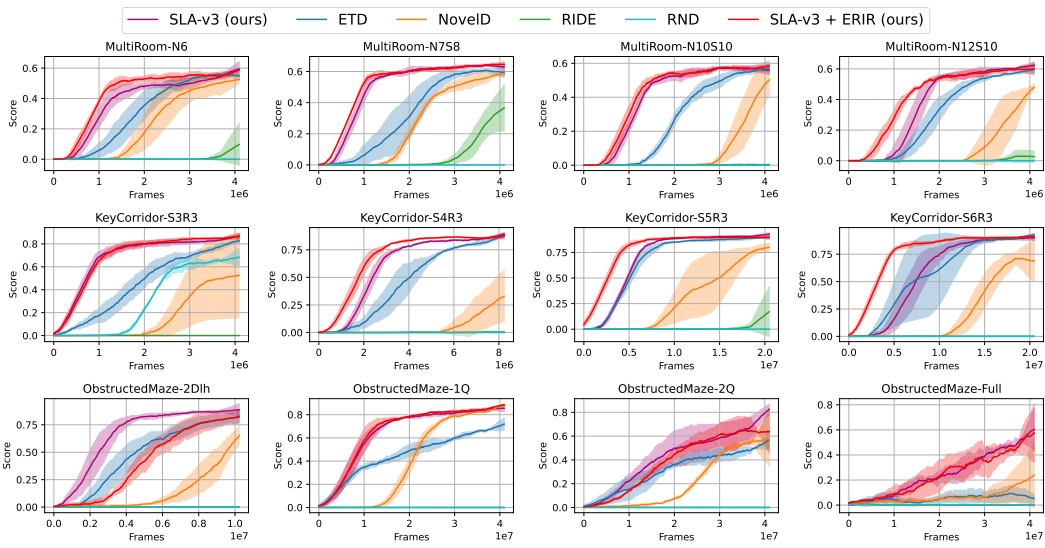

Figure 2: Training performance in the MiniGrid environments. SLA-v3 achieves better performance compared to baselines across most evaluated environments.

reward is defined as:

$$r_{\text{int}}(s_t) = r_{\text{sla}}(s_t, \tau) \cdot \max(b_\phi(s_t), \beta), \tag{5}$$

where $\beta > 0$ guarantees a minimum exploration incentive (we set $\beta = 0.06$ for the main experiments), and $b_\phi$ represents the novelty detector's output.

Second, the novelty detector serves as an alternative heuristic model architecture to accelerate heuristic adaptation. The heuristic model, as defined by Equation 3, exhibits an inherent slow upward adjustment tendency, which can lead to delayed correction of underestimation for previously unseen hard-to-traverse states. By leveraging the novelty detector's architecture, which typically provides high initial value outputs for unseen states, we utilize it to mitigate this slow correction issue and effectively warm up the heuristic.

In this paper, we employ Random Network Distillation (RND) Burda et al. (2018) as our novelty detector. Its output is computed as the $L_2$ norm of the difference between the prediction network and target network outputs: $\|\phi_{\text{pred}}(s) - \phi_{\text{target}}(s)\|_2$.

## 4 EXPERIMENTS

We evaluate the effectiveness of SLA-v3 across two benchmark families: MiniGrid, representing procedurally generated environments, and the challenging Atari suite, specifically PitFall! and Montezuma's Revenge for their extremely hard-exploration characteristics. Our implementation is built upon PPO Schulman et al. (2017) as the foundational reinforcement learning algorithm. To manage intrinsic and extrinsic rewards, we employ separate value functions with distinct discount factors, consistent with prior work Burda et al. (2018); Kazemipour (2020). All environments utilize shaped rewards, defined as $\max(\text{sign}(r), 0)$. All results are averaged across three random seeds, with $1\sigma$ error bars consistently depicted in all figures. More details about the environment and the experiment setting are available in the Appendix.

### 4.1 MINIGRID ENVIRONMENTS

The MiniGrid benchmark Chevalier-Boisvert et al. (2018) consists of procedurally generated grid-world environments. We conducted evaluations on three high-sparsity environments: (1) Multi-Room (MR), (2) Key Corridor (KC), and (3) Obstructed Maze (OM).

In MiniGrid environments, Episodic Restriction on Intrinsic Reward (ERIR Zhang et al. (2021b)) has shown effectiveness by utilizing hash-based episodic visit counts, $\mathbf{I}(N_{eps}(s) = 1)$, to modulate rewards. Building upon this, we also implemented SLA-v3+ERIR, a variant of SLA-v3 with a modified intrinsic reward defined as $r_{int}(\cdot) = [r_{sla}(\cdot) + \alpha\mathbf{I}(N_{eps}(s) = 1)] \cdot \max(b(\cdot), \beta)$, where we fixed $\alpha = 0.05$ for all experiments. Our experiments compare SLA-v3 against four baselines: RND Burda et al. (2018), RIDE Raileanu & Rocktäschel (2020), NovelD Zhang et al. (2021b), and ETD Jiang et al. (2025). It is important to note that a direct comparison with Go-Explore was not feasible due to fundamental incompatibilities with procedurally generated environments, a point further elaborated in Section 2.2. All methods process raw partial observations as inputs for both the policy and intrinsic reward models, reserving full observations exclusively for ERIR computation, as in Zhang et al. (2021b).

**Results**. As depicted in Figure 2, SLA-v3 consistently demonstrates substantial performance advantages over baseline methods, significantly outperforming RND, RIDE, ETD, and NovelD in most scenarios. These findings affirm that the exploration strategy of SLA-v3 effectively generalizes to procedurally generated environments, with its heuristic model exhibiting robust adaptation to novel environment instances sampled from the underlying distribution. While the inclusion of ERIR offers additional performance benefits, our experiments indicate that the fundamental SLA-v3 method alone achieves competent performance, underscoring the inherent effectiveness of its core design.

**Analysis**. Figure 3 illustrates the effectiveness of the *SLA Heuristic* in assigning high reward values to critical states within the KeyCorridor-S6R3 environment. Four key interaction stages—key observation, key acquisition, approaching a locked door with the key, and door unlocking—consistently correspond to monotonically increasing heuristic values. This precisely reflects the prerequisite relationships among these sequential behaviors, confirming SLA-v3's capability to implicitly generate an automatic curriculum. Importantly, even under partial observability constraints, SLA-v3 successfully captures essential information regarding environmental traver-

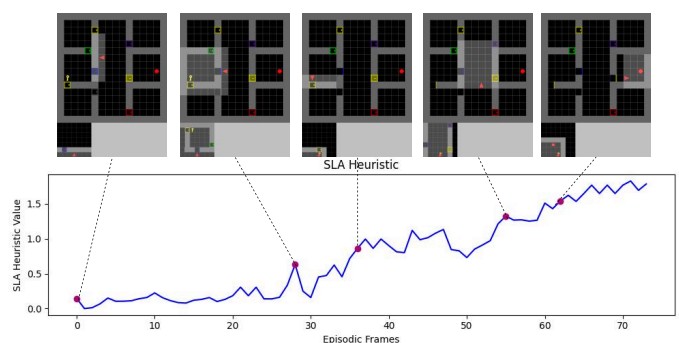

Figure 3: Visualization of the *SLA Heuristic* in the MiniGrid KeyCorridor-S6R3 environment. The monotonically increasing heuristic values for key observation, key acquisition, approaching a locked door with the key, and door unlocking align with their prerequisite relationships, showcasing SLA-v3's implicit automatic curriculum generation.

sal difficulty by effectively leveraging the limited yet informative landmarks available in partial observations.

## 4.2 ABLATION STUDIES IN MINIGRID

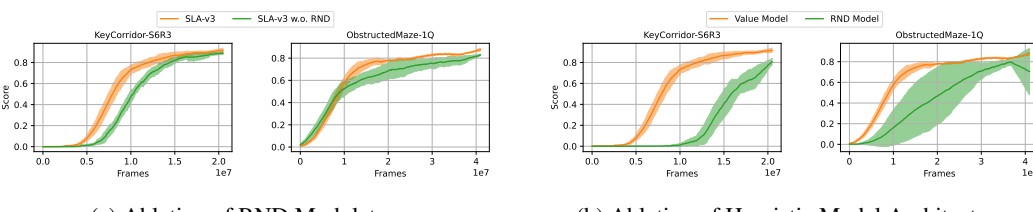

(a) Ablation of RND Modulator.    (b) Ablation of Heuristic Model Architecture.

Figure 4: Ablation study on the integration of RND.

**RND Integration**. Our comprehensive ablation study examined RND integration from two perspectives: its role as a reward modulator and its application in the heuristic model architecture. Figure 4a demonstrates that while the RND modulator benefits SLA-v3's performance, the standalone SLA heuristic achieves comparable results. This is likely due to MiniGrid's depth-focused structure; limited deep branching paths (beyond extrinsic rewards) prevent the SLA heuristic from being trapped in suboptimal, reward-lacking deep branches. Figure 4b further reveals that the RND-based heuristic model architecture underperforms the standard value network architecture. This suggests that in MiniGrid's relatively simple environments, RND's architectural instability may not be fully offset by its heuristic warm-up advantages.

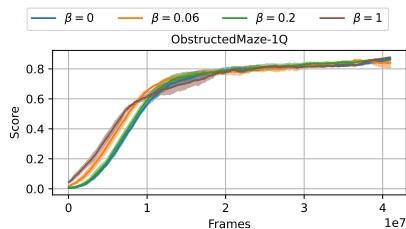

Figure 5: Ablation of RND Modulator parameter $\beta$.

**Hyperparameter Sensitivity**. We also performed an ablation study to examine the sensitivity of key hyperparameter choices: the minimum RND modulator parameter $\beta$ and the downward-biased heuristic loss parameter $\epsilon$. Figure 5 illustrates that the choice of parameter $\beta$ has a limited effect on performance. Figure 6 indicates that the parameter $\epsilon$ should be carefully selected—neither too high nor too low—to achieve an optimal trade-off between overestimation avoidance and heuristic learning speed.

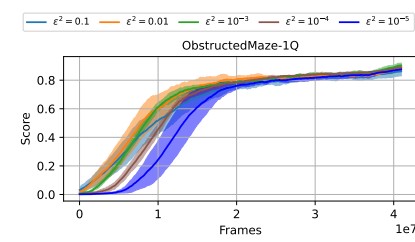

Figure 6: Ablation of heuristic loss parameter $\epsilon$.

### 4.3 CHALLENGING ATARI

We selected the notoriously challenging Atari environments PitFall! and Montezuma's Revenge (MR) to evaluate the depth-focused exploration capability of SLA-v3. PitFall! is characterized by extreme reward sparsity across 255 interconnected rooms, many of which contain hazardous elements that instantly terminate an episode. Successful navigation requires precise, long-horizon action sequences, with a minimum of six consecutive rooms to traverse for even the most accessible extrinsic reward. Historically, only a handful of methods, such as Go-Explore Ecoffet et al. (2021), have successfully solved PitFall!, with most failing to achieve any non-zero extrinsic reward. Montezuma's Revenge presents a similar hard-exploration challenge with three levels, each comprising 24 interconnected rooms with analogous hazardous elements.

Following Go-Explore Ecoffet et al. (2021), we integrate carefully designed domain knowledge for the Pitfall! and the MR environment, utilizing room indices and agent positional information. This knowledge serves two primary purposes: (1) as an input feature for the heuristic model and (2) for pruning sub-optimal branching paths. These modifications establish an optimized testbed for evaluating SLA-v3's sustained deep exploration ability. We employ a value model architecture for

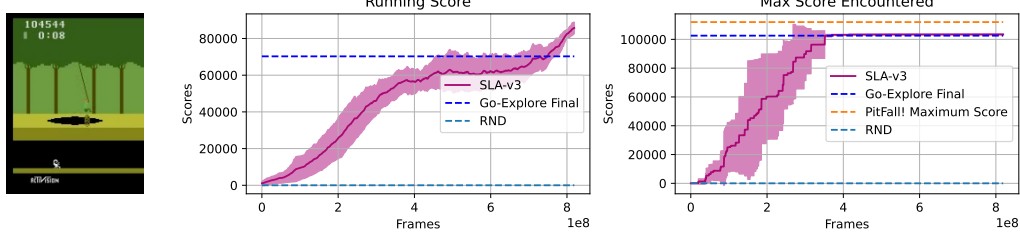

Figure 7: Results for PitFall!. (Left) A rendered view of the PitFall! environment, illustrating SLA-v3 achieving over 100,000 episodic scores. (Middle) Running episodic scores during the rollout process. (Right) Maximum episodic scores encountered during the rollout process.

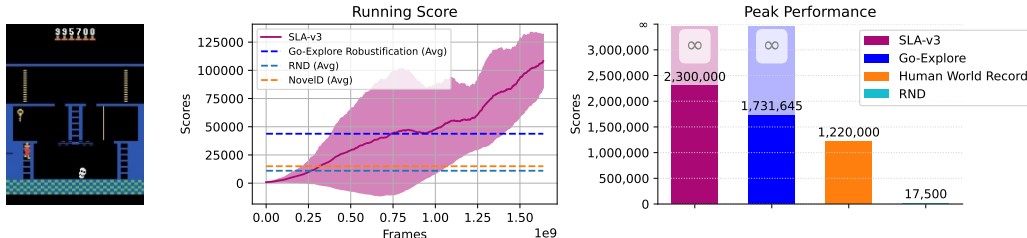

Figure 8: Results for Montezuma's Revenge (MR). (Left) A rendered view of the MR environment, showing SLA-v3 achieving nearly 1 million episodic scores. (Middle) Running episodic scores during the rollout process. (Right) Peak performance of SLA-v3 with greedy action selection and a released episodic step limit. Notably, both SLA-v3 and Go-Explore demonstrate the potential to achieve infinite scores, given the level-3 cyclical property inherent in the MR environment.

our heuristic model. Notably, we omit the RND reward modulator in this setting, as sub-optimal branching paths are already pruned.

**Experimental Results**. As illustrated in Figure 7, SLA-v3 achieves significant performance gains in Pitfall!, attaining an average score of 80,000, with maximum scores exceeding 100,000. These results substantiate SLA-v3's capacity for effective, depth-focused exploration in challenging sparse-reward environments. Furthermore, all three trials occasionally surpassed 100,000 during the roll-out, indicating potential for high-reward discovery and sustained performance. For Montezuma's Revenge, SLA-v3 also achieves average scores over 100,000 during training, though limited by the 18,000-frame episodic step limit. We also applied a simple post-processing method to the trained policy network, using a greedy policy and releasing the episodic step limit (to 180,000 training frames). This increased the maximum score to over 2,300,000, aligning with the maximum scores reported in the Go-Explore paper (1,700,000+). It is important to elaborate on the potential for infinite scores in MR for both SLA-v3 and Go-Explore. The environment is deterministic and features only three level maps. Agents progress from Level 1 to 2, then to 3. Crucially, completing Level 3 restarts the agent at the beginning of Level 3. This cyclical, deterministic design allows for continuous score accumulation once an optimal greedy policy for completing Level 3 is learned. Thus, with the episodic step limit removed and no undiscovered bugs, both SLA-v3 and Go-Explore could theoretically achieve infinite scores.

**Analysis**. To elucidate the challenges of sustained exploration and the advantages of SLA-v3, we examine the Pitfall! Atari environment. As depicted in Figure 9, successful learning in Pitfall! fundamentally involves the sequential traversal of multiple rooms to reach a final objective. For instance, advancing from Room 1 to Room 7 (the yellow objective) via intermediate Rooms 3, 4, and 5 poses significant hurdles. Even assuming consistent visitation of Room 3, reaching Room 5 from Room 4 and subsequently Room 7 requires fulfilling three critical conditions: (1) robust traversal to Room 4 by consistently executing a low-fault-tolerance action sequence through hazardous areas like the tar pit in Room 3; (2) effective exploration within Room 4 to discover and execute precise, low-fault-tolerance actions to bypass quicksand and crocodiles, gaining access to Room 5; and (3) sustained visitation of Room 5 for continued progress toward the ultimate goal.

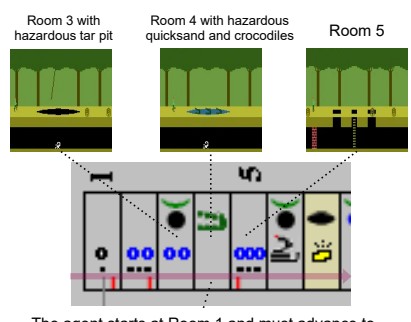

Figure 9: Key rooms in Pitfall! illustrating traversal challenges and derailment risks. A detailed map with explanation is available in Appendix.

The stringent, low-error-tolerance action sequencing inherent in Pitfall!, combined with persistent exploratory noise, collectively generates significant derailment risks that frequently hinder the agent from consistently reaching Room 4 from Room 3. Furthermore,

these challenges, often exacerbated by training-induced policy oscillations, lead to temporal detachment, where the agent loses the ability to reliably return to previously visited critical states such as Room 4. SLA-v3 effectively addresses these challenges by introducing sustainable exploration incentives. Specifically, because Room 4 is located at a higher *shortest-path quasi-metric* from the episodic initial state (Room 1) than Room 3, its SLA heuristic value is inherently higher. Successful visitation of Room 4 from Room 3 therefore increases the maximum observed heuristic value, generating a positive intrinsic reward for the agent. This mechanism fosters persistent advancement toward the exploration frontier, simultaneously demonstrating resilience to derailment and enabling consistent recovery from detachment events.

## 5    DISCUSSIONS

**Robustness to Noisy-TV**. We evaluated the robustness of SLA-v3 in a modified PitFall! environment subjected to uniformly random noise. As depicted in Figure 10, SLA-v3 consistently maintains non-zero rewards even under random noise, albeit with a slight performance degradation. This behavior is attributed to the fact that uniformly random noise lacks landmarks for traversal difficulty and contributes minimally to the heuristic value approximation. Consequently, the relative ordering of state traversal difficulty, and its approximation, is preserved. These results collectively demonstrate that SLA-v3 exhibits robustness to random noise when such noise contains no meaningful information.

**Robustness to Stochasticity**. We also evaluated SLA-v3 in stochastic settings within a modified PitFall! environment, incorporating a sticky-action wrapper. As Figure 10 illustrates, SLA-v3 continues to achieve non-zero extrinsic rewards, despite our shortest-path quasi-metric being inherently defined for deterministic settings. The SLA Heuristic remains effective by providing

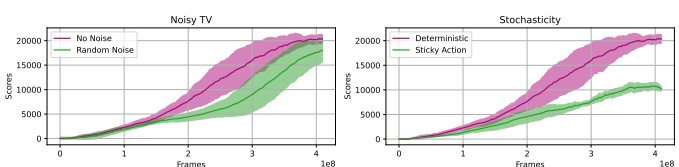

Figure 10: Robustness Analysis of SLA-v3 in PitFall! under Noisy-TV and Stochasticity. (Left) SLA-v3 exhibits resilience to random noise. (Right) SLA-v3 is robust to stochasticity. Success criterion: achieving above-zero episode scores.

high-level exploration incentives, where successful guidance primarily requires maintaining correct ordinal relationships between temporal-abstracted states (e.g., accurately ranking Room 4 as harder to traverse than Room 3 in PitFall!). The observed performance degradation in this stochastic setting is primarily due to the low-fault-tolerance nature of the challenging PitFall! environment, where the agent is not guaranteed to advance to the next room because of potentially incorrect actions taken under the sticky-action mechanism.

## 6    CONCLUSION

This paper introduced a novel paradigm for intrinsic motivation grounded in state traversal difficulty, embodied by our proposed SLA-v3 method. SLA-v3 leverages the shortest-path quasi-metric from $S_0$ as a heuristic for traversal difficulty, thereby systematically promoting the visitation of hard-to-traverse states to mitigate the detachment-derailment problem. Furthermore, SLA-v3 integrates a novelty detector to facilitate heuristic warm-up and prevent stagnation within single trajectories. Extensive experiments conducted in MiniGrid and challenging Atari environments (PitFall! and Montezuma's Revenge) robustly demonstrate the efficacy of SLA-v3.

However, the current SLA Heuristic implicitly assumes that each action step incurs a uniform cost, as it quantifies minimum temporal steps. This assumption constraints its effectiveness in environments where transition costs vary substantially, such as those with intrinsically uncontrollable steps or low-fault-tolerance actions (e.g., navigating from critical "near-death" states versus performing routine actions). Therefore, a promising direction for future work involves modeling the heterogeneous costs of individual steps and incorporating cost-sensitive metrics (e.g., successor distance Jiang et al. (2025); Myers et al. (2024)) into the heuristic target calculation.

# 7 ETHICS STATEMENT

The authors confirm that this work adheres to the ICLR Code of Ethics. This research focuses on advancements in reinforcement learning algorithms and does not involve human subjects, sensitive personal data, or any applications that could lead to direct societal harms. All datasets used are publicly available and appropriately cited. We have ensured that our methodology and findings contribute positively to the scientific community and align with responsible AI development principles. No potential conflicts of interest or sponsorship have influenced the research outcomes.

# 8 REPRODUCIBILITY STATEMENT

To facilitate the reproducibility of our results, we have made every effort to provide comprehensive details regarding our methodology, experimental setup, and implementation. Our novel algorithm, SLA-v3, is thoroughly described in Section 3 of the main paper. For all experimental results, including hyperparameters, training configurations, and environmental specifications, detailed information can be found in Section 4 and Appendix. The code for our proposed methods and experiments, along with instructions for setting up the environments and running the code, is available in the supplementary materials and will be made publicly available upon acceptance.

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
