# OpenReview forum: "SLA-v3: Spatial Linkability-Aware and Novelty-Encouraging State Heuristic for Exploration"
_ICLR.cc/2026/Conference — ICLR 2026 Conference Withdrawn Submission_

### Official Review · Reviewer_sxAA · 2025-10-24

**Soundness:** 2
**Presentation:** 2
**Contribution:** 3
**Rating:** 4
**Confidence:** 3

**Summary:**

The paper presents an intrinsic reward method that encourages the RL agent to visit state which are hard to reach under random exploration. This property of a state is modelled by training a model to predict the timestep corresponding to the first occurrence of a state in an episode. The proposed intrinsic reward then guides the agent towards states that are hard to reach as predicted by this model and once this is achieved, it decreases, by design, to allow for exploration at the frontier of the agents state visitation distribution. Overall, the idea is sound and the motivation to draw inspiration from Go-Explore while attempting to get rid of the assumptions of the latter is relevant. The experiments presented and the evaluation protocols are rigorous and extensive although sometimes not adhering to standards (especially in terms of baselines compared and environment modifications).

**Strengths:**

The combination of the proposed quasi-metric for state reachibility with the design of an intrinsic reward to appropriately maximize it is relevant and well executed. While similar ideas have been proposed in prior work (e.g. Go-Explore), this paper makes a step forward in proposing a method which relies less in specific properties of the environment to achieve good performance (e.g. like the ability to reset the environment to a desired configuration).

Although I have some issues with the formatting of the paper (e.g. citations, wording, clarity), the ideas presented flow nicely and are organized appropriately throghout the paper, allowing the reader to somewhat easily follow. The Figures are well designed and also help. The mathematical principles used to design the reward function are rigorously presented and clear.

The method achieves good performance in MiniGrid and 2 hard-exploration Atari environments (although modified). The experiment protocols are appropriate and the results provide statistically significant conclusions. The authors thoroughly analyzed the behavior of the agents in the environments under different conditions (noise, sticky actions), and learning dynamics under different hyperparameters, which provides valuable insights into the performance of SLA-v3.

**Weaknesses:**

In terms of formatting, the current citation style makes it hard to read the paper. I reocmmend using \citep{} instead of \cite{} unless directly referencing the authors of a paper.

I find this statement unclear: "Intrinsic Motivation [...] enhances the reward signal by leveraging historical experience replay, thereby facilitating more effective policy optimization". IM methods don't have to forcefully use historical experience replay, and hence it is not a good definition of these methods.

Generally, I recognize AI-generated text in several sentences of the paper, which although is not necessary a bad thing, and knowing that authors ackonwledge this in the Appendix, I personally dislike it and find it less clear at times. I would suggest making the effort of writing original text to improve the clarity of the presentation.

The paper omits relevant citations to prior work in using temporal distances in reinforcement learning [1,2] (similarly used in this work to define the quasi-metric for state reachibility), in the state-dependent reachibility properties in MDPs [3,4] (see the line of work in *empowerment* in RL), and intrinsic motivation [5] - only to list a few papers.

The paper does not justify some design decisions of the presented method: e.g., why is RND chosen as the additional source of novelty? why not a different intrinsic reward method?

How and why were the baselines selected? Other open-source intrinsic reward algorithms which have been evaluated in both MiniGrid and Atari environments were omitted (e.g. Disagreement, ICM, E3B, NGU...) [5]. Showing that SLA-v3 outperforms these in standard evaluation settings would help strengthen the claims of the paper a lot.

The Pitfall and MR games from Atari were modified to include additional information (e.g. room index) which can dramatically facilitate the exploration problem. Why weren't the aforementioned baselines not evaluated in these settings as well?

Finally, I believe it would be interesting and very relevant to broaden the evaluation of the algorithm to other popular environments like ProcGen, other hard-exploration Atari games, or Crafter, while comparing the performance of SLA-v3 with currently omitted but highly relevant baselines.

With this, I think the paper presents an interesting method which is rigorously motivated and introduced, but lacks clarity and provides minimal evidence of the value the algorithm could yield (e.g. superior performance compared to well-established baselines). I think improving these 2 points will render the paper publishable.

[1] Park, Seohong, Oleh Rybkin, and Sergey Levine. "Metra: Scalable unsupervised rl with metric-aware abstraction." arXiv preprint arXiv:2310.08887 (2023).

[2] Stooke, Adam, et al. "Decoupling representation learning from reinforcement learning." International conference on machine learning. PMLR, 2021.

[3] Latyshev, A. K., and A. I. Panov. "Skill Learning with Empowerment in Reinforcement Learning." Pattern Recognition and Image Analysis 34.3 (2024): 535-542.

[4] Eysenbach, Benjamin, et al. "Diversity is all you need: Learning skills without a reward function." arXiv preprint arXiv:1802.06070 (2018).

[5] Yuan, Mingqi, et al. "Rlexplore: Accelerating research in intrinsically-motivated reinforcement learning." arXiv preprint arXiv:2405.19548 (2024).

**Questions:**

Why is RND chosen as the additional source of novelty? Why not a different intrinsic reward method?

How and why were the baselines selected? Many open-source, relevant intrinsic motivation baselines, which have been previously evaluated on the environments used in this paper were omitted [1].

[1] Yuan, Mingqi, et al. "Rlexplore: Accelerating research in intrinsically-motivated reinforcement learning." arXiv preprint arXiv:2405.19548 (2024).

---

> ### Author Response · Authors · 2025-11-21
>
> # Response to Reviewer sxAA
>
> We sincerely thank you for your very detailed and constructive review. We are particularly grateful for your positive assessment of our core idea, which you describe as "relevant and well executed," and for acknowledging that our method "makes a step forward in proposing a method which relies less in specific properties of the environment." We also appreciate your recognition of the rigor of our experiments, the clarity of our mathematical principles, and the insightful analysis of SLA-v3's behavior.
>
> Here are our point-by-point responses to your weaknesses and questions:
>
> ## Weaknesses & Questions:
>
> **1. Citation Style:** We apologize for the inconsistent citation style. We will standardize the use of `\citep{}` instead of `\cite{}` to improve readability.
>
> **2. Unclear IM Definition:** You are correct. Our original intent was to convey that IM methods often utilize information stored in a replay buffer, but the phrasing was ambiguous and overly restrictive. We will directly state "using information from the replay buffer" to improve clarity and precision.
>
> **3. Omitted Relevant Citations:** We sincerely thank you for providing these highly relevant and important citations. We recognize that our literature review was not as comprehensive as it should be, particularly in the areas of temporal distances in RL, state-dependent reachability/empowerment, and other intrinsic motivation algorithms. We will incorporate all suggested papers ([1-5] as listed in your review) into the revised manuscript, discussing their connections to our work.
>
> **4. The Choice of RND:**
> Our decision to combine SLA with RND was primarily due to RND being a **well-established and widely-used novelty detector**, often serving as a robust modulator in various successful IM frameworks (e.g., NovelD, MADE, NGU). We acknowledge that exploring combinations with other IM methods is a valuable direction for future work.
>
> **5. Baseline Selection & Modified Atari Environments:** You raise valid points about our baseline selection and the evaluation in modified Atari settings.
>
> * **Baseline Selection Rationale:**
>   * **For hard-exploration Atari (Montezuma's Revenge and Pitfall!):** Our primary comparison was with Go-Explore, given its uniquely impressive, outlier performance compared to all other methods in these extremely challenging tasks. Our goal was to demonstrate that SLA-v3 can achieve comparable depth exploration capability to this milestone method. To provide broader context and address potential concerns about baseline scope, we also included other representative intrinsic motivation baselines.
>   * **For MiniGrid:** Our baseline selection was guided by influential prior work in this domain (e.g., NovelD), and we also incorporated recent state-of-the-art methods like ETD to ensure our evaluation remains aligned with the latest advancements.
> * **Omitted Baselines:**
>   * Some methods, like ICM, have been shown to perform poorly (e.g., zero scores) in MiniGrid environments, as noted in prior work (e.g., the NovelD paper). Including such baselines would not significantly strengthen our claims regarding SLA-v3's efficacy in these specific tasks.
>   * For others, like NGU, a lack of official open-source code and the significant engineering effort required for reliable and fair reimplementation (given the subtle details often critical for their performance) make their inclusion beyond the scope of a single paper.
> * **Modified Atari Environments & Go-Explore Comparison:** The PF and MR environments were modified (adding room index, agent position) to enable effective state representation, mirroring techniques used in Go-Explore. Our evaluation in these settings primarily aimed to demonstrate that SLA-v3 can achieve **competent performance comparable to the impressive final results of Go-Explore**, thus showcasing our algorithm's depth exploration capability in such challenging conditions. Re-evaluating an exhaustive set of baselines in these *specific modified settings* would constitute a very substantial research project, requiring immense computational resources (single runs often take 7 days).
>
> **6. Broadening Evaluation:** We appreciate your suggestion to broaden the evaluation to environments like ProcGen, other hard-exploration Atari games, or Crafter. Our current work focused on two distinct objectives: performance in **procedurally generated environments** (MiniGrid) and in **extremely hard exploration tasks** (Montezuma's Revenge, Pitfall!). The selected baselines already effectively support our claims within these defined scopes. Expanding to a wider array of environments and a more exhaustive comparison against a broader set of IM baselines are indeed excellent and important directions for **future work**.
>
> We hope these clarifications comprehensively address your concerns and enhance the understanding of our work.

---

### Official Review · Reviewer_zKas · 2025-10-31

**Soundness:** 3
**Presentation:** 2
**Contribution:** 3
**Rating:** 2
**Confidence:** 3

**Summary:**

The paper shows that an intrinsic reward based on a model of the time distance from the start can be very effective on some really sparse-reward games such as Montezuma's revenge and Pitfall.

**Strengths:**

- very nice introduction and related work section

- excellent experimental results on difficult RL problems: in particular on Montezumas Revenge and Pitfall games

- the proposed intrinsic reward defined in Eq. (4) is interesting: you get rewarded for states that are further away from the start than all other states along the trajectory, so practically avoiding going backwards.

- Eq. (5): combines the intrinsic reward of Eq. (4) with novelty detector (here Random Network Distillation, RND).  Basically, the intrinsic reward from Eq. (4) modulates the intrinsic reward from a novelty detector.

**Weaknesses:**

- the "shortest-path quasi-metric" suggests that there is some graph-theoretic shortest path involved, however, the heuristic just learns the time index.  don't get me wrong, I like simple ideas and solutions, but then why call it "shortest-path quasi-metric"?  I see that you use it to motivate your idea, the SLA heuristic, but then you do not use it at all (and instead just the time index).

- you only explain the intrinsic reward that you want to use (Eq. (5)), but then how do you create the whole RL algorithm with it.  That part is completely missing!

- also what model/network are you using for $H_θ$?

- figure 1 is confusing:  is it generated?  the overlay grayed-out neural network makes no sense and looks like an indicator for AI-generated content.

- you didn't mention whether and how you used LLMs for writing the paper!

**Questions:**

- the title: Spatial Linkability-Aware and Novelty Encouraging State Heuristic.  Why so complicated!  Aren't you just encouraging to explore states far away from the start?  What is linkability-awareness?

- Figure 1, left: the easy-to-traverse states are close to the start, the hard-to-traverse states are far away.  Aren't there many many more hard-to-traverse states in high dimensions?  Or are you using some special properties of the state topology, that there are not so many hard-to-traverse states?

- Figure 1, middle: what are the boxes in the pink box?  why a box with "Architecture Choice"?  What do the arrows mean?  The middle panel looks automatically generated with an LLM (e.g., the grayed-out overlayed neural net).  It doesn't make much sense to me!

- Figure 1, right: similar, is this just a generated figure?  I find it confusing!  Why the "Agent" blue discs on the arrows?  Why converging arrows to "Novelty?  Looks all fancy, but does it really mean anything?

- Figure 1, caption: also the caption isn't a real sentence.  There is an extra verb "has been reached".???

- line 201: you write that your agent switches from **go** to **explore**?  when?  how?  you don't show us the whole algorithm!

- is the shortest-path quasi-metric a "quasi-metric"?  what are the requirements to be a quasi-metric?

- You define in Eq. (1) the SLA Heuristic.  Where do you use it?

- For the SLA Heuristic, you have to define equality of states.  Are you using the equality on pixalated images (like GoExplore)?  Or how do you do that?  E.g. in Montezuma's revenge there is a skull going left and right, so two states are rarely exactly the same...

- Line 180: "the ground truth shortest-path quasi-metric serves as an up-
per bound t".  Isn't that the other way around?

- Also, on the RHS in Eq. (1) you write $s_t$.  But how is that variable bounded?  What is it?  Does it range over all possible states of all possible trajectories you have seen so far?  Or over all theoretically possible trajectories?  The $\min$ should specify what it is ranging over.

- what is $H_θ$ in Eq. (2)?  I know it is your heuristic model, but you do not mention  that connection in the text.

- line 193: what is "functionally equivalent"?  you mean the value model in PPO has the same (type) signature as the heuristic model?  Or do they compute similar things?

- line 196: why is no neural architecture search necessary?  Maybe learning $H_θ$ requires a different architecture than the value model of PPO.

- Figure 7: do have the curve of Go-Explore?  you just show that you reach and exceed the "Go-Explore Final" score.  But are you reaching it faster?

---

> ### Author Response · Authors · 2025-11-21
>
> # Response to Reviewer zKas
>
> Thank you for your exceptionally detailed and critical review. Your feedback highlights crucial areas for clarity and completeness, particularly regarding methodology, terminology, and presentation. We value your insights, which are essential for improving our work.
>
> We appreciate your recognition of our "excellent experimental results" and the "interesting" nature of our intrinsic reward. We are committed to addressing your concerns with clearer explanations and enhanced precision.
>
> Here are our point-by-point responses:
>
> ## 1. Weaknesses & Questions on Terminology and Concepts
>
> **1.1 "Shortest-path quasi-metric" Terminology (Weakness 1, Question 7):** Our "shortest-path" concept represents the minimum steps from $s_0$ in a directed graph. This inherently forms a quasi-metric. While a "distance" function typically implies symmetry ($d(s_1,s_2) = d(s_2,s_1)$), a **quasi-metric allows for asymmetry** ($d(s_1,s_2) \neq d(s_2,s_1)$). Since the minimum steps from $s_0$ to a state can be unequal to the minimum steps *to* $s_0$ from that state, the term quasi-metric is correctly applied here.
>
> **1.2 Title Complexity & "Linkability-Awareness" (Question 1):** "Linkability-awareness" refers to the SLA heuristic's ability to identify states that are topologically "further" or "less connected" from the start. These states are then sparsely-covered under random exploration. Our heuristic guides exploration into these sparsely linked regions.
>
> **1.3 SLA Heuristic Usage & Variable Definitions (Questions 8, 11, 12):** We apologize for the ambiguity. Eq. (1) defines the *ground truth* $H_{sla}(s)$ over all theoretically possible trajectories, which is ideal but intractable. Instead, we learn a neural network approximation, **$H_{\theta}(s)$**, as an approximation of $H_{sla}(s)$, which we will clarify in the paper. The $t$ in Eq. (1) represents the length of a theoretical trajectory $\tau$ from $S_0$ to $s'$ (i.e., $t = |\tau_{S_0 \to s'}|$ for the minimum path).
>
> **1.4 "Upper Bound" Correction (Question 10):** You are correct. The observed time **$t$** serves as an **upper bound** for the true minimum number of steps (**$H_{sla}(s)$**) required to reach that state, as the true minimum cannot exceed any observed path length.
>
> **1.5 "Functionally Equivalent" (Question 13):** By "functionally equivalent," we mean that both the heuristic model and the value model map from the state space to a scalar space, effectively sharing the same input/output space structure.
>
> **1.6 "Agent switches from go to explore" (Question 6, Line 201):**
> This refers to a **conceptual behavioral mode** dynamically emerging from our $r_{sla}$ reward (Eq. 4), not a discrete algorithmic switch. The agent receives intrinsic reward only when $H_{\theta}(s_t)$ for the current state $s_t$ is greater than the maximum $H_{\theta}$ value observed in the trajectory so far. This actively pushes the agent towards further states ("go"). Once the agent reaches states where it can no longer surpass this maximal $H_{\theta}$ value within a trajectory, the $r_{sla}$ reward will diminish to zero. At this point, the agent transitions to an "explore" mode, driven primarily by policy entropy, to diversify actions in those deep regions.
>
> ## 2. Weaknesses & Questions on Methodology and Algorithm Details
>
> **2.1 Missing Complete RL Algorithm (Weakness 2):** We respectfully disagree that this part is "completely missing." Eq. (4) and Eq. (5) define the intrinsic reward, and Section 4.1 specifies our use of a PPO-based framework with separate intrinsic/extrinsic value models. While these details are present, we acknowledge the need for a single, comprehensive algorithm overview.
>
> **2.2 $H_{\theta}$ Model Architecture (Weakness 3, Question 14):** As noted in the paper, $H_{\theta}$'s architecture is designed to match the task's value model or RND model. Regarding NAS (Question 14), NAS was not necessary as our chosen architectures proved empirically sufficient and computationally efficient, balancing performance with resource constraints.
>
> **2.3 State Equality Definition (Question 9):** This is an excellent and insightful question. For complex Atari games like Montezuma's Revenge and Pitfall!, we leverage **game-specific abstract state representations** (e.g., room indices, agent positional information) for robust state identification, following Go-Explore, as mentioned in Section 4.3.
>
> **Due to the strict character limit of this rebuttal system, our comprehensive response to your detailed review has been divided into two parts. Please refer to "Response to Reviewer zKas - Part 2" for the remainder of our answers.**

---

> > ### Author Response · Authors · 2025-11-21
> >
> > Response to Reviewer zKas - Part 2
> >
> > This document serves as the continuation of our response to Reviewer zKas's detailed feedback, addressing the remaining points from your comprehensive review.
> >
> > ## 3. Weaknesses & Questions on Figure Presentation
> >
> > **3.1 Figure 1 Clarity (Weakness 4, Questions 2, 3, 4, 5):**
> > We recognize that Figure 1 was found confusing and appeared "AI-generated," indicating that its current presentation did not effectively convey our intended message. We aim to clarify the purpose and information within this figure to address these concerns.
> >
> > * **Left Panel:** This panel was intended to conceptually illustrate that states "further" from the start, as perceived by our SLA heuristic, are generally harder to traverse under random exploration. It serves as an abstract representation to convey this difficulty distribution rather than literally mapping all states.
> > * **Middle Panel:** This panel aimed to show the flexibility of $H_{\theta}$'s architecture. Specifically, "Architecture Choice" indicated that $H_{\theta}$'s neural network can match (or draw inspiration from) the value model or RND model used in the task, leveraging existing structural designs.
> > * **Right Panel:** This panel intended to metaphorically depict the combined guidance of SLA (encouraging deeper exploration away from the start) and RND (driving discovery of novel states). The elements, including blue discs representing the agent and converging arrows towards "Novelty," were symbolic representations of this dual exploratory drive.
> >
> > ## 4. Weaknesses & Questions on Writing and Experiment Presentation
> >
> > **4.1 LLM Usage (Weakness 5):** We **DID** mention the LLM usage in the supplementary materials (appendix), where LLMs were only used for language polishing and guidance on figure layout.
> >
> > **4.2 Figure 7 Go-Explore Curve (Question 15):** We acknowledge that including the full Go-Explore learning curve would provide a more complete comparison of learning speed. However, our primary goal in Figure 7 is to demonstrate that SLA-v3 achieves **competent performance comparable to the impressive final results of Go-Explore** on these challenging tasks. This effectively showcases SLA-v3's exceptional capability for deep exploration in environments where Go-Explore also excels, focusing on **ultimate exploration capacity** rather than the entire learning trajectory.
> >
> > We hope these clarifications comprehensively address your concerns and enhance the understanding of our work.

---

### Official Review · Reviewer_hA8N · 2025-11-01

**Soundness:** 2
**Presentation:** 3
**Contribution:** 2
**Rating:** 4
**Confidence:** 4

**Summary:**

This paper tackles the Detachment-Derailment (D-D) problem in sparse-reward reinforcement learning. The authors argue that the root cause of D-D lies in the environment's topological transition structure, where some states are inherently difficult to traverse and revisit.

They propose SLA-v3, a novel intrinsic motivation (IM) framework centered on a "state traversal difficulty" heuristic. This heuristic, H_{sla}, is defined as the shortest-path quasi-metric (minimum number of steps) from the initial state S_0. The method learns to approximate this metric by training a heuristic model H_{\theta} to predict the minimum episodic timestamp (step count) for each state, using a downward-biased loss. The intrinsic reward encourages visiting states with higher heuristic values than those seen so far in the episode. This is combined with a novelty detector (RND) to prevent stagnation.

The method is evaluated on MiniGrid and the challenging Atari environments (PitFall!, Montezuma's Revenge), demonstrating strong performance against IM baselines.

**Strengths:**

1. **Clear Problem Motivation:** The paper provides a clear definition of the Detachment-Derailment (D-D) problem, grounding it in the environment's topological structure.
2. **Novel Heuristic:** The core idea of using the shortest-path quasi-metric from the initial state as a heuristic for "traversal difficulty" is novel, intuitive, and a sensible approach to tackling the D-D problem.
3. **Practical Approximation:** The method for learning this heuristic by approximating the minimum episodic timestamp from a replay buffer is a practical and clever approach. The downward-biased loss (Eq. 2) is well-suited for this minimum-seeking objective.
4. **Strong MiniGrid Analysis:** The method shows compelling performance in procedurally generated MiniGrid environments. The visualization in Figure 3 provides strong evidence that the learned heuristic captures a meaningful representation of task progress, showing monotonically increasing values at key sequential stages (e.g., key acquisition, door unlocking).

**Weaknesses:**

1. **Mismatch between Heuristic and Stochastic Environments:** The core SLA heuristic (H_{sla}(s) = d(s_0, s)) is defined as the shortest path, a concept well-defined in deterministic settings. However, the paper evaluates this in stochastic environments (e.g., sticky-action PitFall) and claims it "remains effective". The theoretical justification for this is weak. In a stochastic environment, a state might be reachable in 10 steps (with 1% probability) and 50 steps (with 99% probability). The heuristic's loss function (Eq. 2) is designed to optimize towards the minimum (10 steps). It is unclear why this optimistic, minimum-seeking objective is the most appropriate or stable choice for a stochastic setting, as opposed to an expected path length or a more risk-aware metric.*
2. **Reliance on Domain Knowledge in Atari:** The paper rightly argues against Go-Explore's dependency on state restoration, which limits its applicability. However, the impressive results on PitFall! and Montezuma's Revenge also "integrate carefully designed domain knowledge," specifically "room indices and agent positional information". This reliance on privileged information significantly weakens the claim of a general, domain-agnostic solution and makes the comparison to other methods less direct. It raises the question of how much of the exploration challenge is being solved by this domain knowledge versus the heuristic itself.
3. **Incomplete Literature Review and Missing Citations:** The related works section (2.1), while covering the main paradigms, omits several significant and foundational papers in the exploration space. This incomplete review of the literature weakens the paper's positioning and fails to properly contrast the proposed method with other established approaches. Specifically, the following key citations are missing:
   * **Foundational Novelty/Diversity Methods:**
     * Tang et al. (2017). "# exploration: A study of count-based exploration for deep reinforcement learning." Advances in neural information processing systems 30.
     * Hong et al. (2018). "Diversity-driven exploration strategy for deep reinforcement learning." Advances in neural information processing systems 31.
   * **Alternative Exploration Frameworks:**
     * Jin et al. (2020). "Reward-free exploration for reinforcement learning." International Conference on Machine Learning. PMLR.
   * **Alternative State Metrics:**
     * Wang et al. (2023). "Efficient potential-based exploration in reinforcement learning using inverse dynamic bisimylation metric." Advances in Neural Information Processing Systems 36.
   * **Recent Related Work:**
     * Hao et al. (2025). "Llm-explorer: A plug-in reinforcement learning policy exploration enhancement driven by large language models." Advances in Neural Information Processing Systems 2025.

**Questions:**

* (Re: Weakness 2) How critical is the domain knowledge (room indices, agent position) to the performance in the PitFall! and Montezuma's Revenge experiments? How does SLA-v3 perform compared to baselines (e.g., RND) if this domain knowledge is removed and all methods must learn from pixels alone?
 * (Re: Weakness 1) The heuristic H_{\theta} is trained to approximate the minimum observed timestamp. In a highly stochastic environment, this objective seems optimistic. Could the authors comment on how this minimum-seeking objective affects policy learning and stability compared to a potential average-seeking objective (e.g., approximating the mean timestamp)?
 * The method combines a PPO-based RL algorithm (with separate value functions), the SLA heuristic model H_{\theta} (trained with Eq. 2), and a novelty detector (RND, which has its own loss). Could the authors please provide the exact formula for the overall optimization objective used in the implementation, showing how all these different loss components are combined and weighted?

---

> ### Author Response · Authors · 2025-11-20
>
> # Response to Reviewer hA8N
>
> We sincerely thank you for your thorough review and insightful comments on our submission. Your feedback is highly valuable and will greatly help us improve the quality of our paper. We appreciate your positive remarks on the clear problem motivation, novel heuristic, practical approximation, and strong MiniGrid analysis.
>
> We have carefully considered all your points and address your weaknesses and questions below:
>
> ## 1. Mismatch between Heuristic and Stochastic Environments (Weakness 1 & Question 2)
>
> We acknowledge that the core SLA heuristic is conceptually clearer in deterministic settings. However, its effectiveness in stochastic environments stems from its ability to capture the **macro-level relative exploration difficulty, or value ordering,** among states, rather than providing a perfectly precise shortest-path prediction. In sparse-reward settings, what agents primarily need is a guiding signal that indicates the direction of deeper exploration, not an exact timestamp.
> Our downward-biased loss (Eq. 2) guides the agent towards less-visited, deeper regions by approximating the **minimum observed timestamp**. Using an average-seeking objective could overestimate traversal difficulty for frequently revisited states, yielding a less effective signal. The RND component further ensures robust exploration.
>
> ## 2. Reliance on Domain Knowledge in Atari (Weakness 2 & Question 1)
>
> We agree on the observation regarding domain knowledge (room indices, agent position) in our Atari experiments, which we transparently disclose.
>
> * **Comparison to Go-Explore:** Go-Explore's impressive results on challenging Atari games similarly relied on prior knowledge or complex pre-processing. Specifically, the auxiliary information used in our experiments, such as room indices and agent positional data, was adopted directly from the Go-Explore framework. Our primary goal in these highly challenging environments was to demonstrate the upper bound of SLA-v3's exploration capabilities under comparable experimental conditions, thereby proving the competitiveness of our core exploration mechanism.
>
> * **Challenges with Raw Pixel Inputs & High Experimental Cost:** For Montezuma's Revenge and PitFall!, extracting effective macroscopic exploration signals directly from raw pixel inputs is extremely difficult due to high-dimensional temporal noise. Furthermore, the computational expense for these experiments is substantial; a single run often takes up to 7 days, making the development of a fully robust raw-pixel method prohibitively costly.
>
> Therefore, to effectively evaluate and highlight the contribution of our SLA heuristic method without being overwhelmed by the additional complexities of raw pixel inputs in these specific, exceptionally hard environments, we adopted proven auxiliary information, allowing for fair comparison with SOTA techniques. Future work will focus on robust representation learning to address raw pixel challenges and enhance generality.
>
> ## 3. Incomplete Literature Review and Missing Citations (Weakness 3)
>
> We appreciate your valuable suggestions for improving our literature review. We will incorporate your feedback in the revised version.
> * **Intrinsic Motivation (IM) & State Metrics:** We will add the foundational novelty/diversity methods and the alternative metric method to our references to provide a more comprehensive overview of intrinsic motivation research.
> * **Beyond IM Scope:** Methods like LLM-Explorer (Hao et al.), which emphasize LLM-based action space shaping, fall outside the scope of our paper as we focus on a purely intrinsic motivation framework without external knowledge-driven planning or action space intervention. We included Go-Explore in our discussions not as an IM method, but because its core concept of efficient exploration and replay provided inspiration for our intrinsic motivation approach. We will clarify this distinction in the revised paper.
>
> ## 4. Overall Optimization Objective (Question 3)
>
> To clarify the overall optimization objective used in our implementation, we can highlight two key aspects:
>
> *   **Intrinsic Reward Definition:** We define an intrinsic reward within a framework similar to RND, where intrinsic and extrinsic value functions are kept separate. We considered this a standard practice in intrinsic motivation literature, hence not explicitly detailed in the main text.
> *   **Loss Function Combination:** All loss components—for the PPO policy, the SLA heuristic model $H_{\theta}$, and the RND novelty detector—are summed together, each with a uniform weight of 1. This approach avoids introducing additional hyperparameters for balancing the losses, simplifying the optimization process.
>
> We hope these clarifications provide a clearer understanding of our implementation details.

---

> > ### Comment · Reviewer_hA8N · 2025-11-25
> >
> > Thanks for the detailed response. I appreciate the honesty regarding the computational constraints (7 days/run) and the difficulty of learning from raw pixels in the Atari experiments.
> >
> > However, this admission effectively confirms my primary concern. By stating that extracting signals from raw pixels is "extremely difficult," you are acknowledging that the method currently relies on privileged information (room indices, coordinates) to function in complex environments. While the comparison to Go-Explore is valid in that specific context, I believe a general exploration method targeting ICLR should ideally demonstrate efficacy without such hand-crafted features to prove robustness.
> >
> > Regarding the stochasticity issue, the explanation about "macro-level relative exploration difficulty" is intuitive, but I am still missing concrete evidence (like an ablation comparing minimum vs. average objectives) to prove this minimum-seeking heuristic is stable in general stochastic settings.
> >
> > Given these remaining limitations on generality and robustness, I will be maintaining my score.

---

### Official Review · Reviewer_zxCd · 2025-11-03

**Soundness:** 3
**Presentation:** 3
**Contribution:** 3
**Rating:** 6
**Confidence:** 3

**Summary:**

This paper tackles the Detachment-Derailment problem in sparse-reward exploration. The proposed solution, SLA-v3, utilizes a traversal difficulty heuristic (shortest-path distance from $S_0$) to incentivize visiting these difficult states. This method achieves strong results on hard benchmarks (MiniGrid, Atari) without relying on state restoration.

**Strengths:**

- Strong performance on challenging hard-exploration benchmarks
- intuitive design (heuristics)
- clearly written, well-motivated

**Weaknesses:**

- The paper's primary motivation is to overcome Go-Explore's reliance on state restoration (teleportation). However, the Go-Explore paper itself introduced a "policy-based" (goal-conditioned) variant specifically to address this limitation, which also does not require state restoration. This paper appears to overlook this highly relevant baseline.
- The central argument against state restoration is its impracticality in real-world applications, with robotics being the most prominent example. The paper's claims of practical applicability would be  strengthened by demonstrating SLA-v3's effectiveness in such a domain (e.g., solving Franka Kitchen-gym tasks or other manipulation/locomotion challenges; hard-exploration robotics benchmark).

**Questions:**

None

---

> ### Author Response · Authors · 2025-11-14
>
> # Response to Reviewer zxCd
>
> Thank you for your positive feedback on our work's intuitive design and strong performance, and for your constructive comments. We appreciate the opportunity to clarify the motivations regarding the policy-based Go-Explore and the benchmark selection.
>
> ## The Inapplicability of Policy-based Go-Explore in Procedurally Generated Environments
>
> You raise an excellent point regarding the policy-based variant of Go-Explore. We are indeed familiar with this work and acknowledge its contribution in reducing the dependency on explicit state restoration in *singleton environments*.
>
> However, the core challenge we address arises in **procedurally generated environments**, where a fundamental constraint makes *any* form of Go-Explore inapplicable: each call to the env.reset() function generates a completely new, unique environment instance with different initial states, layouts, and dynamics. The **Policy-based Go-Explore** variant, while not needing explicit resets, relies on reaching goal states identified from a past instance. In a procedurally generated setting, a goal state (or its representation) discovered in one instance is almost certainly **irrelevant, invalid, or unreachable** in a new instance. The goal-conditioned policy thus lacks a reliable target for cross-episode exploration.
>
> ## Regarding the Benchmark Selection
>
> We sincerely thank you for the suggestion to validate our method on robotics benchmarks like Franka Kitchen. We agree that demonstrating success in such domains would be a strong testament to an algorithm's practical utility.
>
> The primary objectives of this work are twofold: to advance exploration in (a) procedurally generated environments and (b) known yet extremely hard exploration domains. Our experimental design directly reflects this dual focus. The **MiniGrid benchmark** was chosen specifically to validate the first objective in environments where state recovery is fundamentally impossible. Concurrently, the **Atari hard-exploration benchmarks (e.g., Montezuma's Revenge, Pitfall)** were included for the second objective to demonstrate the deep exploration capability against such challenging environments. We believe this benchmark selection comprehensively covers the core claims of our paper.

---

### Note · Authors · 2026-01-15

I have read and agree with the venue's withdrawal policy on behalf of myself and my co-authors.